# Factors influencing walking trips. Evidence from Gdynia, Poland

**Marcin Wolek** [ID] [*], **Michal Suchanek** [ID] [*], **Tomasz Czuba** [ID] [*]

Faculty of Economics, University of Gdansk, Gdansk, Poland

☉ These authors contributed equally to this work.
* marcin.wolek@ug.edu.pl

**Data Availability Statement:** The data base used in this research could be directly accessed at: https://osf.io/bhskr/quickfiles.

**Funding:** This research did not receive any specific grant from funding agencies in the public, commercial, or not-for-profit sectors.

## Abstract

Political support for active mobility is growing for many reasons, including land use planning, health, and improved mobility. As the vital part of many cities is their central area, decision-makers need to know what factors are essential for increasing walkability. This paper aims to identify the main factors affecting the walkability of the city centre of Gdynia (Poland). To achieve this, the research design was adjusted to the specificity of the local use case. Based on primary data collected via personal interviews, factor analysis was applied to rule out potential collinearity and reduce dimensions. Logistic regression models were then constructed. The results were compared with the research carried out in other cities. The results show that only two of the analysed factors are significant, namely accessibility and safety. Both are extensive categories and include many subcomponents that are influential among different groups of citizens. Our research also confirms that walkability is a city-specific issue that is influenced by many local factors.

## 1. Introduction

### 1.1. Need for the support of active modes

For centuries the cities have been designed from a pedestrian perspective. Only the development of public transport enabled their spatial expansion, furthered by individual motorisation development.

As the car remains the dominant mode of transport for most European cities, the adverse external effects of transportation are concentrated in these urban areas [1]. Problems such as air pollution, noise, congestion, and reduced quality of life are far from being solved [2]. Transport is responsible for a quarter of the E.U.'s greenhouse gas emissions and still growing [3].

The response to the negative consequences of car-dependent cities is a concept of sustainable urban mobility [4]. It manifests itself in the close link between transport and land-use planning. The European Commission recommends Sustainable Urban Mobility Plans (SUMP) as a helpful tool for planning and implementing transport policies in cities [5–7]. An

**Competing interests:** The authors have declared that no competing interests exist.

essential element of sustainable urban mobility planning focuses on active modes, including walking [8], [9]. It is one of the most influential and investment-efficient measures linking sustainable mobility, health improvement, and urban society's well-being [10–13].

The advantages of active modes include increased mobility and accessibility, reduced dependence on the passenger car, and reduced demand for parking spaces [14] and improved safety for all transport users. Improving people's health using urban travel modes is also highlighted [15–17], as walking has a dual nature. On the one hand, it is an expression of daily mobility or part of it. On the other hand, walking is a way of spending free time and the desired form of physical activity.

Implementing complex measures on sustainable mobility needs a precise diagnosis of challenges and barriers. As many measures are focused on active mobility, it is essential to know their role in the mobility market at a local level. Therefore, this paper aims to identify the main factors affecting the city centre's walkability in Gdynia (Poland).

The research on walking mobility has many methodological and organisational challenges. The authors have attempted to confront the results discussed in this article with the research carried out in other cities. It has led us to state the main research question (RQ1): What are the main factors affecting the probability of taking a walking trip to the city centre in Gdynia (Poland)?

### 1.2. Structure of the paper

The paper discusses the main factors affecting the probability of taking a walking trip to the city centre in Gdynia. The structure of the paper is, therefore, as follows. The Introduction (chapter 1) provides background for further analysis and includes the research question. Chapter 2 contains an overview of walkability in the literature. In section 3, the study site (city of Gdynia) was presented with a particular focus on its central area. Also, the marketing research process was described with an initial data presentation. Results of marketing research data analysis form a backbone of section 4 supported by the discussion. Chapter 5 (Conclusions) summarises the whole paper and provides additional remarks on the study's limitations.

While designing the study, the authors were aware of the difficulties in comparing pedestrian surveys' results. They result, among others, from different methodologies used in conducting the research and how contact was made with the respondent and the area, analysed or the period in which the research was conducted. Moreover, specific environmental and societal factors are critical for such comparisons.

### 2. Literature review

There are various reasons for the increased interest of researchers in walking. Apart from being a core part of transport and mobility, health and leisure activities, walking plays a vital role in determining cities' vitality and liveability [18]. For many years walking has been treated as such a common, usual activity that sometimes it was not considered a means of transport [11]. The underestimated scale of pedestrian mobility is shown by Barcelona's example, where walking accounted for 73% of all journeys lasting less than 10 minutes [19].

Currently, there are many walking research methods. Their variety is a reflection of the complex nature of walking in the urban environment. These methods include field observation, questionnaires (use of validated measures and self-report), G.I.S. analyses, web applications and social media data mining [20].

## 2.1. The concept of walkability

As there are many factors influencing pedestrian traffic, its evaluation is methodologically challenging. Hence, the walkability concept's popularity flexibly captures the most critical issues at the crossroads between urban infrastructure and functions and the individual user.

It is a response to the complexity of pedestrian traffic, which takes into account the purpose of the trip (transport or leisure), its environmental [21] and health benefits [13], especially for certain groups of citizens. Walkability was one of the most critical factors determining subjective well-being [22].

The spatial dimension of walkability includes different spatial factors related to cities' organisation and functionality [23]. A more narrow approach defines walkability as "*a measure of whether the built environment of a neighbourhood encourages people to walk*" [24]. It underlines the importance of the built environment on walking in urban areas.

Other definitions stress the efficiency and pleasure of walking [25], and the measurement of how friendly an area is to walking.

The concept of walkability can also be found in such terms as "walkable environments" [19], "walkable streets", and "walkable places" [26]. The latter is related to walking-sized areas of mixed land-use [27]. Walkability is also a central concept in T.O.D. (Transit-Oriented Development) policies as walking accessibility is crucial for high-performance public transport in urban areas of high density [28–30]

## 2.2. Factors determining walking in urban areas

The majority of factors influencing walking fall into two main categories: environmental factors resulting from an urban form and socio-economic features of the users [31], including cultural characteristics of the walker [32]. When gender is taken into account, "*socio-economic issues appear to be more relevant than infrastructure development in explaining gender differences in commuting patterns*" [33].

Other features are the influence on other members' travel choices and preferences [34–36], individual characteristics of a given person and household, and season and weather characteristics [37–40]

Urban form and neighbourhood design factors influence walks, but their length could depend on individual socio-economic factors. In Montreal's specific case, the research focused on elderly citizens showed decreasing walking distance with seniors' increasing age [41].

Although multi-use urban structure and design are prerequisites for sustainable urban growth, some conflict between achieving a high-density urban space of desired quality may be found. These can be social, i.e. lack of privacy, competition for space between different users [42], environmental [43], including impact on the micro-climate [44]. An analysis of residents in 14 cities showed that the frequency of walking could decrease above a certain level of urban density, although thresholds differ according to the type of walking trip [45]. This is in line with the results of a study, which found that the built environment showed a stronger association with walking than general physical activity [46].

The impact of the built environment on walking activity can be expressed in the development of 3D concept (design, density, diversity) [47] into 5D concept (adding destination and distance) [48] and further—into 7D (adding demographics and demand management) [22, 49]. The 7D concept is not only of theoretical nature—it has been used to evaluate how urban design qualities impact walkability in the central part of Dallas, US [50].

The built environment is especially relevant for active modes, as they are more (directly) "*exposed to the surroundings compared to car and public transport users*" [37]. Determinants that shape walking activity, in this case, are land-use diversity, intersection density [49],

density [51], urban structure [52] and information [50]. Also, the type of area surveyed could influence the walking distance [53].

### 2.3. Literature review summary

The precise measurement of walking brings complex challenges, and the results are strongly influenced by the research method [54]. Although there is no standard definition of a walking trip, its importance for sustainable urban mobility and, in a broader perspective, for the sustainable development of urban areas is indisputable [55].

The literature review indicates numerous factors influencing walking. They could be divided into two main groups: socio-economic factors, depending on the individual features of citizens and households (i.e. age, sex, health status, car-dependence) and the built environment [56]. Walkability defines the conditions created by the built environment. The latter even varies depending on the specific city (centre, vast suburbs). At the crossroads between these two groups, the perception of a given space and its evaluation occurs.

## 3. Method

In the first stage, a goal was formulated to which the scope of the study was subordinated. Due to the need to obtain detailed information, it was decided to use the direct interview method (face to face), which allowed for high-quality data to be obtained from respondents. The development of the questionnaire and the implementation of field research were made possible thanks to cooperation with the self-governmental unit Road and Greenery Administration in Gdynia. Factor analysis was carried out after processing the collected data. It was applied to rule out potential collinearity and reduce dimensions. Logistic regression models were then constructed to verify the effect of the independent variables on the probability of a person taking a walking trip to the city centre of Gdynia (Fig 1).

### 3.1. Study site

Gdynia is a mid-size city located in northern Poland on the Baltic Sea. It was founded and developed thanks to the port's construction in the 1920s and 1930s and granted city rights in 1926 before quickly becoming one of the most dynamically developing cities in the Baltic Sea Region.

It has a population of 246,000 and together with Gdansk, Sopot, and other communes form the Metropolitan Area of Gdansk-Gdynia-Sopot, being one of the most important urbanised areas in Poland.

The spatial layout of Gdynia could be described as island-linear, with half of the area covered with forest (Fig 2). A specific feature of the urban space is the seaport's central location, which borders the city centre. The central part of a city is formed by the compact historical quarter buildings created in the inter-war period and new building complexes of high intensity with various functions [57].

The central district of Gdynia creates a unique architectural composition at the junction of land and sea in the city centre. The spatial scope of research has covered a significant part of the central district (Śródmieście), where every twentieth citizen of Gdynia officially lives. The population density is lower than the whole city (Table 1), but the area under analysis is the destination of many journeys related to work, education and leisure. More than 1/3 of the inhabitants of the central district are older people.

The centre of Gdynia is characterised by a high level of public transport service, which in this part of the city is based primarily on zero-emission trolleybuses [60]. Accessibility to rail

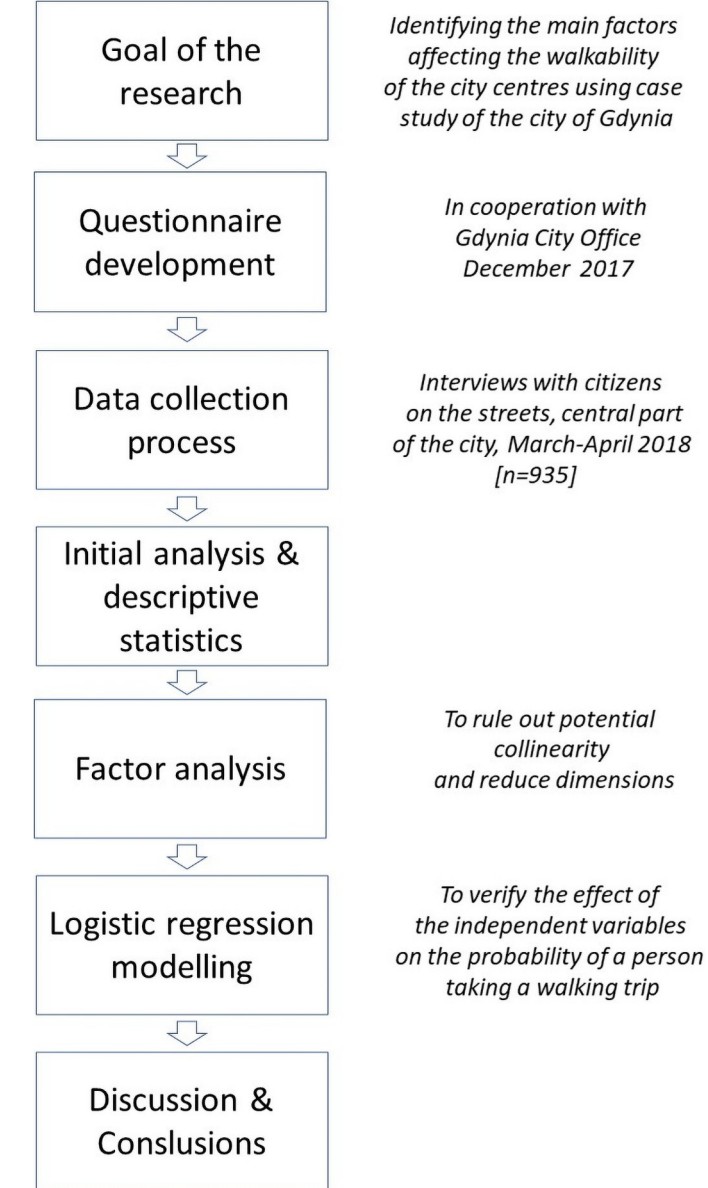

**Fig 1. The research process.** *Source: self-study.*

transport is ensured by the railway station, which in 2018 served 14.9 million passengers (7th place in Poland) [61]. The high accessibility of public transport is why the percentage of households with access to a passenger car is lower than in the whole city (Table 2).

In Gdynia and other Polish cities, the motorisation index has grown substantially during the economic transition post-1989. It was accelerated with E.U. accession in 2004 and, nowadays is, together with suburbanisation, the most critical challenge for sustainable urban mobility planning [5]. The motorisation index in Gdynia amounted to 602 cars per 1000 inhabitants in 2018 and was 84% higher than in 2004. Gdynia is not an exception, and this trend was and is a characteristic feature of the whole country (Fig 3).

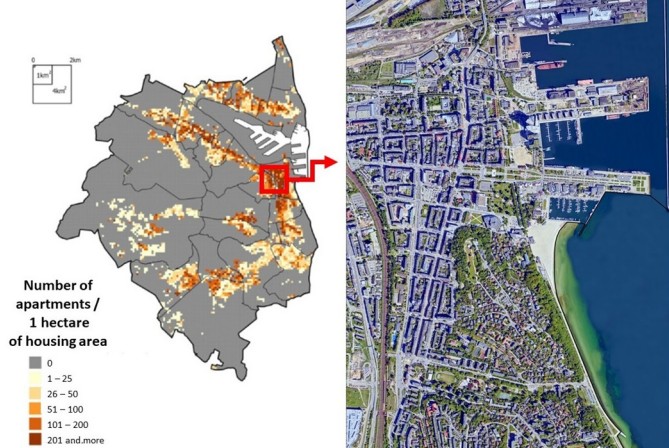

**Fig 2. City of Gdynia and its central area as the place of the research.** *Source: self-study based on the* [57] *and OpenStreetMap.*

## 3.2. Walkability in Gdynia in a strategic context

One of the goals of the Strategy of Gdynia City Development adopted in 2017 is "friendly public space in Gdynia's districts". It includes, among other things, limiting the dominant role of cars in the city by giving priority to walking, cycling mobility and public transport [63] (Table 2).

This was the first study of this kind in the city of Gdynia. Previous studies focused on evaluation of pedestrian traffic in the city center. Following the 2013 research results obtained via observation, 1,200 to 2,000 pedestrians per hour were recorded on the researched area's main streets [64]. Research conducted in 2014 in this area on pedestrian traffic as part of the CIVITAS DYN@MO project showed that its share in the modal split is comparable to other European cities and amounts to between 23% and 24%, a typical working day [65].

## 3.3. Data collection process

The marketing research was carried out as part of the European Union's FLOW ("Furthering Less Congestion by creating Opportunities for more Walking and cycling") project. The ZDiZ ZDiZ—Zarząd Dróg i Zieleni, Road and Greenery Authority, part of self-government administration of the city of Gdynia organised a procedure to select a contractor for the study. The contractor completed the research and delivered the initial data set to the ZDiZ. The study was

**Table 1. Basic features of the central district and the whole city of Gdynia.**

| Feature | Unit | Gdynia | Central district |
|---|---|---|---|
| Population | inhabitant | 246 348 | 11 549 |
| Area | km² | 135 | 11.49 |
| Density | inhabitant/km² | 1823 | 1005 |
| The population of post-working age | % | 26 | 35.6 |
| Orography | n.a. | hilly | flat |
| Forests | % of the total area | 45.9 | 0 |
| Households with at least one car (2018) | % | 75.5 | 67.9 |

*Source: self-study based on* [58, 59]

**Table 2. Walking in the strategic documents of the city of Gdynia.**

| Strategic document | Year of adoption | Relevance to walking |
|---|---|---|
| Sustainable Energy Action Plan (SEAP) | 2015 | Supporting transport development strategies meeting citizens' needs, developing cycling, walking, and improving the road infrastructure, including technical and organisational measures. |
| Sustainable Urban Mobility Plan | 2016 | A need to improve the quality of urban space, especially in the city center |
| | | Operational goals: 1.1. Improving conditions for pedestrian traffic [actions: Improving the quality and consistency of walking routes, Traffic priority for pedestrians, Improvement of walking safety], 1.4. Improving the quality of public space [i.e., improving space quality in the central district leading to traffic reduction]. |
| The Strategy of Gdynia Development | 2017 | Directions of actions: |
| | | Limiting the dominant role of cars in the city by shaping urban space, taking into account the priority role of walking and cycling mobility and public transport. |
| | | Adapting Gdynia's urban space for pedestrians' needs, including families with children and the elderly and disabled people. |
| | | Creation of quiet and restricted traffic zones in the city center and districts or pedestrian zones, with priority for pedestrian and bicycle traffic [. . .] |
| | | Reducing car traffic over short and medium distances by promoting mobility walking and cycling. |
| | | Introducing solutions to reduce traffic in the city, increasing pedestrian safety and cyclists, and creating pedestrian traffic axes. |
| | | Creating—based on small trade and service zones—public spaces in Gdynia's districts, taking into account the priority share of pedestrian and cycling traffic. |
| The General Spatial Master Plan | 2019 | The main objective of transport policy in Gdynia should be to implement the sustainable development by creating conditions for the efficient and safe transport of people and goods with priority given to walking and cycling, public transport and reducing the environmental impact of transport. |
| | | Transport and land use directions of development: The development of facilities for walking, including for people with reduced mobility. |
| Plan of the Sustainable Development of Public Transport for the City of Gdynia for 2016–2025 | 2019 | Improving the accessibility of the public transport system for pedestrians |
| Accessibility Standards for the City of Gdynia. | 2012 | Design of public spaces should take into account the priority given to walking in urban traffic. |

*Source*: *self-study based on* [57, 63, 64]

authorised by the ZDiZ Gdynia. The research was anonymous and did not include sensitive data and issues. To sum up, there was no ethics committee nor an institutional review board, but in our opinion, it was not needed. The questionnaire for the interview is attached to the submission of our paper. All data were fully anonymised at the stage of the data collection process. Apart from measuring the pedestrian traffic volume (observation method), the survey

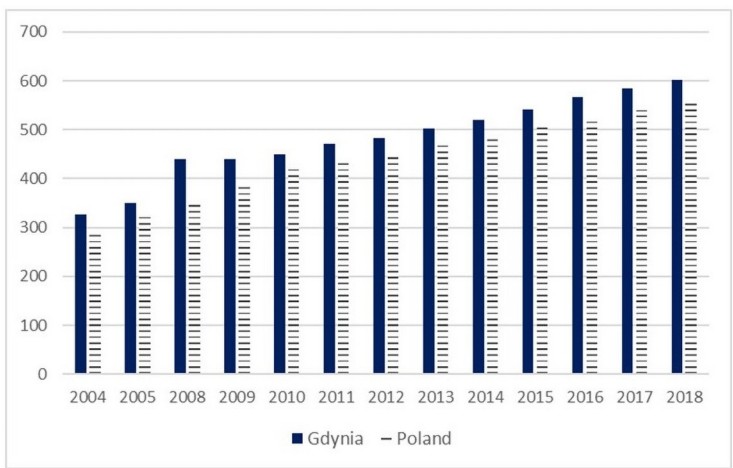

**Fig 3. Motorisation rate index in Gdynia and Poland in 2004–2018 [passenger cars /1000 inhabitants].** *Source*: *self-study on* [62].

was a part of comprehensive traffic research carried within the project "Furthering Less Congestion by creating Opportunities for more Walking and cycling" (FLOW).

The survey questionnaire consisted of twenty-three questions, of which fifteen were related to walkability and eight related to demographic data. The questions related to walkability consisted of single and multiple-choice questions, questions on a 1–5 scale and open-ended questions. The questionnaire also included a separate section on weather conditions during the survey.

The data collection process took place within 14 days (all weekdays including weekends), at different times of the day (morning, early afternoon, afternoon, evening) and in areas with different pedestrian traffic. At each chosen research point in the central part of the city (Fig 2), two interviewers recruited pedestrian for a face-to-face interview. Every sixth pedestrian was asked to answer questions included in the questionnaire. Tourists and persons staying temporarily (at the time of the survey) in Gdynia were excluded from the study through filtering questions asked at the beginning. Each day, interviewers noted the weather conditions included wind, cloudiness, precipitation and air temperature. Weather conditions were recorded on each day during the data collection period but they had little impact on pedestrian traffic. Therefore it was assumed that weather conditions had no substantial impact on the research. In such conditions, the final sample size obtained 934 interviews. The participant expressed his consent to participate in the study orally, and the participation itself was not obligatory. If someone did not want to take part in the survey, he/she could simply refuse. People under the age of 18 (like all participants in the study) participated freely on the same basis—age verification based on the respondent's oral declaration.

54% of respondents were female and 46% male. One in 20 respondents to the survey was under the age of 18., The share of respondents aged 19–34 and 35–54 was similar (respectively 37% and 30%). One in four respondents was aged 55–74. Respondents aged more than 74 consisted of 3% of the sample. More than half of the respondents (57%) had at least one car in the household, 51% worked professionally, and 24% were students. Among the respondents, 63% have at least one child. 35% of respondents declared that the origin of their journey is home. For 16%, the origin was shopping, 9% recreation, 8% work, 7% health, 6% school, meal, public services, and 7% other sources). The answers regarding the journey's destination were different; 23% of respondents declared shopping, 19% home, 13% recreation, 9% work, 8% health

and 7% public services. For 27% of respondents, their journey took up to 10 minutes, for 43% from 10 to 30 minutes. 20% of respondents declared more than 30 minutes, 10% were unable to estimate the duration time (Table 3).

## 4. Results and discussion

The respondents declared whether their walk in the city centre was connected with running errands or whether they were walking for the sake of walking. As the walkability of an area is determined by people's willingness to walk there without additional incentives, it has been decided to treat this declaration as a binary dependent variable (1 –"I have arrived in the city centre to have a walk here"; 0 –"I have arrived in the city centre due to other obligations").

It has been assumed that an area's walkability is strongly affected by the perception and appraisal of different qualities of the built environment, along with factors such as comfort, aesthetics, and safety. People were asked to assess these aspects on a five-point Likert scale (1-poor, 5 –excellent). The results are presented in Fig 4.

Due to the potential collinearity of the analysed variables, factor analysis was applied to rule out potential collinearity and reduce dimensions. This is a method which is advised in order to analyse the structure of the variability of the data. It verifies which variables behave in a similar fashion which indicates that a certain set of variables is correlated. In case of our research, this would indicate that certain aspects of walkability can be grouped together as they are perceived as a certain whole by the respondent. Despite there being a number of researches connected with walkability, we opted for an exploratory factor analysis (EFA) instead of confirmatory factor analysis (CFA), so as no to have to make any assumptions *a priori* regarding the structure, as this could lead to a risk of losing some local specifics of walkability in the process. Given the fact, that no comparable research has been done to date regarding these local specifics, we decided to choose EFA as a more open-ended options. Principal component analysis was adopted as the method of choice for the EFA, as there were no specific prerequisites connected with the type of data. In line, with the Kaiser-Meyer-Olkin criterion, eigenvalues higher than 1.0 were accepted for further research. As a result, five factors that correspond with the theoretical determinants of walkability were identified. It is difficult to find a widely agreed consensus as to the set of factors determining walkability, but the results presented in Table 4 are pretty similar to the findings of Singh [66].

### 4.1. Initial assessment of factors affecting walkability

All of the 23 variables could be aggregated into five factors, representing underlying latent variables—the factors—affecting the walkability as perceived by the respondent. Every factor represents a certain area. If the variables wound up in a certain factor (had a factor loading lower than -0.55 or higher than 0.55 for that given factors) it means that the responses in regards to those questions and they perceived effect on the walkability was relatively similar for the respondents. Each of the factors thus represents a specific aspect of this perception.

The first factor groups together the accessibility and comfort of walking through an area. This includes variables such as movement comfort, accessibility of public transport and accessibility of services.

Comfort is a pedestrian's ability to travel from one point to another without being overburdened by the built environment's inconvenient elements. In other studies, a street network is a crucial element of pedestrian-friendly cities [67] and the built environment's attractiveness [68].

Also, street connectivity was among the most critical factors associated with "walking-for-transport-outcomes" in a complex study involving 17 cities in 12 countries [69].

**Table 3. Basic characteristics of the respondents.**

| Category | Result | |
|---|---|---|
| Sex | Female | 54% |
| | Male | 46% |
| | **Total** | **100%** |
| Age | Under 18 | 5% |
| | 19–34 | 37% |
| | 35–54 | 30% |
| | 55–74 | 25% |
| | More than 74 | 3% |
| | **Total** | **100%** |
| Car in household | No | 43% |
| | Yes—1 car | 42% |
| | Yes—2 cars | 13% |
| | 3 and more cars | 2% |
| | **Total** | **100%** |
| Occupation | Work professionally | 42% |
| | Pensioner | 24% |
| | Studying | 15% |
| | Work professionally and studying | 9% |
| | Does not work | 7% |
| | Maternity leave | 4% |
| | **Total** | **100%** |
| Children in the household | No | 37% |
| | Yes—1 child 30%; | 30% |
| | Yes– 2 children | 26% |
| | Yes– 3 children and more | 7% |
| | **Total** | **100%** |
| Journey origin | Home | 35% |
| | Shopping | 16% |
| | Recreation | 9% |
| | Work | 8% |
| | Health | 7% |
| | School | 6% |
| | Meal | 6% |
| | Public services | 6% |
| | Other | 7% |
| | **Total** | **100%** |
| Journey destination | Shopping | 23% |
| | Home | 19% |
| | Recreation | 13% |
| | Work | 9% |
| | Health | 8% |
| | Meal | 8% |
| | Public services | 7% |
| | School | 4% |
| | Other | 9% |
| | **Total** | **100%** |

(*Continued*)

**Table 3.** (Continued)

| Category | Result | | |
|---|---|---|---|
| Walking time | Up to 10 minutes | 27% |
| | 10–30 minutes | 43% |
| | More than 30 minutes | 20% |
| | I do not know | 10% |
| | **Total** | **100%** |

Source: self-study based on the research carried in Gdynia (n = 934)

The variables grouped in the second factor describe the respondents' perception of green areas and small infrastructure. In particular, three variables were analysed and are grouped in this factor—the perception of benches, including their placement and comfortability, the number of green areas as perceived by the respondent and the number and size of trees and bushes providing shade and creating an atmosphere. The number and size of bushes were a positive factor supporting active transportation [70–72] as it promotes healthy lifestyles. Moreover, in the research of public parks in cities in the U.S., park quantity (measured as the percentage of city area covered by public parks) was one of the strongest predictors for citizens' overall well-being [73]. Recent research on older citizens found evidence of an association between pedestrians' infrastructure and aesthetics with physical function [74] and access to green areas [75].

The relationship between walking and green infrastructure is manifested by the dual nature of this form of activity. On the one hand, it improves the quality of travel to work; on the other hand, it encourages people to spend more time walking for leisure [76].

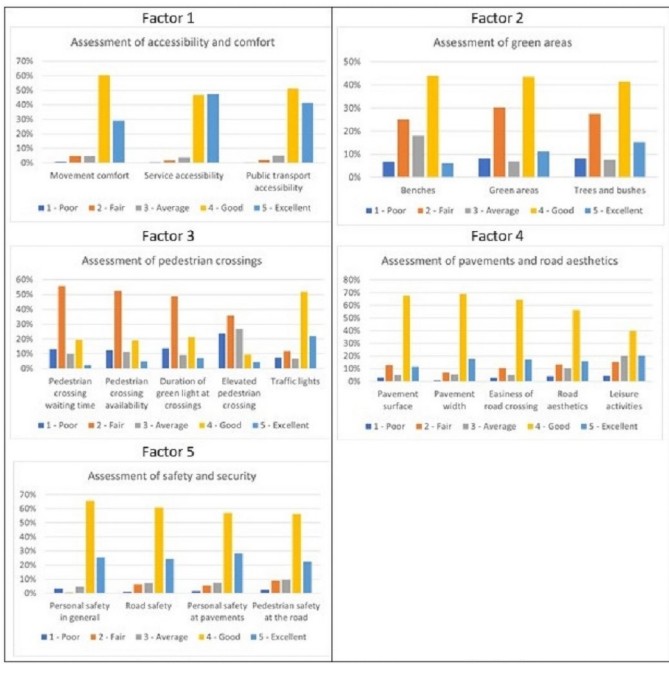

**Fig 4. Perception and appraisal of different parameters of the built environment in the city of Gdynia.** *Source: self-study based on the research carried in Gdynia.*

**Table 4. Results of the factor analysis—Factor loadings for five possible variables affecting walkability.**

|  | FACTOR 1 | FACTOR 2 | FACTOR 3 | FACTOR 4 | FACTOR 5 |
|---|---|---|---|---|---|
| PERSONAL SECURITY IN GENERAL | 0.40 | 0.08 | 0.12 | 0.03 | **0.66** |
| ROAD SAFETY | 0.37 | 0.14 | 0.17 | 0.15 | **0.65** |
| MOVEMENT COMFORT | **0.57** | 0.21 | 0.07 | 0.21 | 0.38 |
| SERVICE ACCESSIBILITY | **0.75** | 0.10 | 0.05 | 0.09 | 0.19 |
| PUBLIC TRANSPORT ACCESSIBILITY | **0.80** | -0.04 | 0.06 | 0.07 | 0.08 |
| BENCHES | 0.15 | **0.64** | 0.17 | 0.10 | 0.16 |
| GREEN AREAS | 0.00 | **0.91** | 0.03 | 0.14 | 0.03 |
| TREES AND BUSHES | 0.04 | **0.90** | 0.01 | 0.13 | 0.01 |
| PEDESTRIAN CROSSING WAITING TIME | -0.02 | -0.10 | **-0.78** | -0.05 | -0.20 |
| PEDESTRIAN CROSSING AVAILABILITY | 0.01 | -0.12 | **-0.69** | 0.00 | -0.14 |
| DURATION OF GREEN LIGHT AT CROSSINGS | -0.03 | -0.10 | **-0.76** | -0.06 | -0.23 |
| ELEVATED PEDESTRIAN CROSSING | -0.18 | 0.11 | **-0.63** | -0.19 | 0.07 |
| PEDESTRIAN CROSSING SAFETY | 0.06 | -0.30 | -0.38 | -0.08 | 0.43 |
| TRAFFIC LIGHTS | -0.30 | 0.17 | **-0.55** | -0.20 | 0.28 |
| PAVEMENT SURFACE | 0.08 | 0.08 | 0.04 | **0.76** | 0.19 |
| PAVEMENT WIDTH | 0.02 | 0.14 | 0.06 | **0.66** | 0.19 |
| PERSONAL TRAFFIC SAFETY AT PAVEMENTS | 0.08 | 0.01 | 0.13 | 0.49 | **0.62** |
| EASINESS OF ROAD CROSSING | 0.02 | 0.14 | 0.20 | **0.55** | 0.49 |
| PEDESTRIAN TRAFFIC SAFETY AT THE ROAD | 0.04 | 0.01 | 0.19 | 0.47 | **0.65** |
| ROAD AESTHETICS | 0.17 | 0.21 | 0.07 | **0.76** | 0.07 |
| LEISURE ACTIVITIES | 0.12 | 0.36 | 0.07 | **0.63** | 0.04 |

* Significant factor loadings in bold

Source: self-study based on the research carried in Gdynia

[77] showed the importance of environmental variables in explaining walking route choices, including visual aspects of the urban landscape. The latter is essential as the pedestrians' perception depends on the sensory experience of walking in the urban environment and influences walking speed [78]. In this case, the importance of clear information/signage for pedestrians could be an essential element of complex infrastructure development dedicated to pedestrians [79]. It could be implemented within the smart city concept, as smart transportation is an important element needed to convert city into a smart city [80].

The third factor groups all the variables connected with the assessment of pedestrian crossings. Interestingly enough, it turned out that all the variables connected with the pedestrian crossing behave in a similar fashion—if the respondent assessed one of the aspects positively, it was quite likely that he assessed all of the positively. These aspects include: the waiting time, green light phases, traffic lights and safety aspects. In general, all of these characteristics are perceived rather poorly by the respondents—only 2% of respondents perceive pedestrian crossing waiting time as excellent. Also, only 4% perceive the pedestrian crossing availability as excellent, whereas 53% perceive it as fair and more than 12% as low. The duration of green lights and elevated pedestrian crossings are perceived only slightly better. The factor analysis indicates that the variability of all these factors' perception is similar, which could mean that they are generally unfriendly to specific groups of people and may negatively affect walkability.

Granié et al. found that crucial factors explaining pedestrians' decision to use particular crossing are the buildings' presence and function, the quality of pavements, and the marked parking spaces [81]. Other studies show that older people are at risk when crossing the street

as they have a reduced walking speed [82]. Similar results were obtained for Cape Town, South Africa [83], Spain [84] and Dublin [85], although in the latter case, the width of the road was also significant.

The fourth-factor groups together different variables connected with pavements, including their surface, width, and aesthetics. Interestingly enough, this factor also includes the accessibility of leisure activities. Interestingly enough, accessibility of leisure activities turned out to be grouped together with the characteristics of the pavements and their quality. This might indicate that the fact that the pedestrian can participate in leisure activities during the trip further improves his perception of the quality of pavement itself. Gdynia may still be perceived as a city in development, with even parts of the city centre still being developed or transformed. During this process, several investments have been made, which a view to increasing the quality of infrastructure in highly frequented areas. This might mean that better quality pavements also characterise the areas with strongly developed leisure services. The accessibility to leisure is perceived better than other factors, with 40% of respondents describing it well and over 20% as excellent. People using these services are also interested in the quality of infrastructure that allows them to reach their destinations. Put together, this creates a vital factor of walkability, as proven by previous research, including Porto Alegre (Brasil), where pavement quality was found to be one of the three most essential walkability attributes [86]. Also, the quality of specific components of the walking infrastructure, including lighting, pavements and street block size, corresponded to higher rates of walking [18, 87]

The fifth factor shows the traffic safety of the pedestrians on the pavement and on the road itself. It shows a different variability than the pavement qualities as such, which are presented in the previous factor. People feel safe in Gdynia, although the safety, in general, is perceived to be slightly better than the safety of pavements (91% of respondents feel good or excellent in terms of their general personal security, whereas 85% do so on the pavements). The traffic safety on the road is the worst factor, with 3% feeling that safety is low and 9% that it is fair, but this still might be regarded as an acceptable result in urban policy. Factor analysis groups general traffic safety with safety on the roads and pavements, which indicates that these two ideas are highly correlated.

Traffic safety and perception of personal security are other crucial factors determining walkability. According to a comprehensive review of [88], the perception of safety and connectivity are the most influential factors in developing a walkable environment. Following the research of (Alfonzo 2005), the fundamental preconditions for walking are safety and comfort conditions, but they are followed by widely defined "pleasurability" [89, 90] that also covers urban greenery and furniture [91]. Further study by [91] confirmed the importance of safety in the case of recreational and destination walking. Furthermore, a survey of parents of schoolchildren in California found that it is not easy to identify a single key factor influencing walking; somewhat, walking behaviour is influenced by a combination of different built environment parameters.

The perception of traffic safety depends on the individual characteristics of citizens. Results of several studies show that older people are at risk when crossing the street as they have a reduced walking speed [82]. Similar results were obtained for Cape Town, South Africa [83], Spain [84] and Dublin [85], although in the latter case, the width of the road was also significant. The above confirms that the perception of safety influences the evaluation of time and distance of a walking journey [79]. Many studies indicate that "safety & security" is one of the most decisive categories affecting every walking trip type [92–94]. This is naturally an extensive category, including safety, which is usually treated as a characteristic of traffic, and security, which is perceived personally and includes danger concerning crime and physical violence.

A more detailed look reveals several essential elements shaping pedestrian safety, including the impact of motorised traffic [95], on-street parking and the existence of cycling infrastructure [96]. In some research, especially in Latin America, traffic safety is analysed together with perceived personal security. Arellana et al. found that the subjective perception of security & traffic safety were the most critical factors influencing walkability in Latin America's cities [97]. It corresponds with Porto Alegre (Brazil) findings, where traffic safety and public security were again two of the three most important attributes of walkability [86].

All of the above factors were treated as independent variables along with a number of socio-economic variables, including gender, age, number of cars in the household, the dominant mode of transport and professional status.

## 4.2. Logistic regression model

Logistic regression models were then constructed to verify the independent variables' effect on the probability of a person taking a walking trip to the city centre. Logistic regression has been widely used to analyse walkability determinants and is a generally accepted method in similar research. The significance of the variables was analysed using the one-by-one approach and in combination with other independent variables. A pseudo-R-squared was calculated to determine the models' explanatory power, and the Akaike Information Criterion (A.I.C.) was used to compare the models' relative quality. The final model presents all the significant independent variables and their effect on walkability. The significant variables include three covariates (factor 1, factor 5 and no. of household cars) and one factor with six different possible responses (professional status). A p-value lower than 0.05 means that the given variable significantly affects the response—the dependent variable (Table 5). This, in turn, means that, for example, a higher value of the accessibility as assessed by the respondent results in a higher chance of him taking the walking trip in the analysed neighbourhood.

## 4.3. Discussion

In answering the research question (What are the main factors affecting the probability of taking a walking trip to the city centre in Gdynia?), only two of the analysed factors were significant: accessibility and safety. The factors representing pavements, pedestrian crossings and green areas were not statistically significant. An increase in perceived accessibility and safety increases pedestrians' chance of taking a walking trip through the city centre.

**Table 5. Results of the optimal logistic regression model—Binary dependent variable (1—Walking trip, 0—Mixed trip).**

|  | LEVEL | VALUE | SE | P-VALUE |
|---|---|---|---|---|
| **CONST** |  | 0.01 | 0.16 | 0.96 |
| **ACCESSIBILITY (FACTOR 1)** |  | 0.14 | 0.06 | 0.03 |
| **SAFETY (FACTOR 5)** |  | 0.15 | 0.06 | 0.01 |
| **NO. OF CARS IN HOUSEHOLD** |  | -0.23 | 0.09 | 0.01 |
| **PROFESSIONAL STATUS** | Unemployed | 0.17 | 0.23 | 0.48 |
| **PROFESSIONAL STATUS** | Employed | -0.53 | 0.17 | 0.00 |
| **PROFESSIONAL STATUS** | Student | -0.26 | 0.19 | 0.18 |
| **PROFESSIONAL STATUS** | Employed student | -0.59 | 0.22 | 0.01 |
| **PROFESSIONAL STATUS** | Pensioner | 0.22 | 0.18 | 0.21 |
| **PROFESSIONAL STATUS** | Parental leave | 0.76 | 0.29 | 0.01 |

Source: self-study based on the research carried in Gdynia

The lack of statistical significance of walking infrastructure like pavements and crossings results probably from the central part of the city's relatively higher infrastructural development.

According to the relative insignificance of the "green factor", the reasons explaining this result include the transport purpose of pedestrian traffic and the fact that there is a large amount of urban greenery (but not forests) in the centre of Gdynia (ca. 30%) as well as the city's proximity to the sea.

An additional car in a household decreases the chance of a person taking a walking trip through the city centre by 23%. Regardless of the complex relationship between car dependence and intensity of land use, it could be said that urban sprawl reinforces the role of the car and diminishes active modes (walking, cycling) and public transport. On the contrary, higher rates of non-motorised trips are characteristic of residents of the areas with high density and diversified land use mix [98].

The professional status also statistically significantly affects the possibility of a person walking just for the sake of walking. People who are unemployed, pensioners or are on parental leave are far more likely to walk just for the sake of it than students or the employed. In most countries, public investments have been concentrated on prioritising infrastructure for cars [99]. However, findings show that in many countries, walking remains the primary mode of transport even today. This is especially true for lower and middle-income groups, especially in situations connected with care-related tasks performed by women. Calculations made for Santiago (Chile) indicate that walking has a share of 56% to 77% in the modal split [89].

Gender, age and the dominant mode of transport were not statistically significant predictors of walkability in this research. The difference in walking activity by gender was found in Brisbane (Australia), where men spent more time walking for transport than women [100]. The opposite results were obtained in the recent study covering several Polish cities [101]. Similar results were reported in Bogotá, where the share of walking trips in the modal split was highest among working women [102].

This research differs from much that has been published so far because it concerned one separate area and not the whole city. The centre of Gdynia is the final destination of many trips associated with work and education, shopping and free time. Each city presents different spatial, economic and social morphology. Therefore substantial variations in walkability indicators are found within the same city [103].

This study's different results confirm that walkability is a city-specific issue influenced by many local factors. Walkability can be treated as a complex set of capacities embodied in any urban morphology [104]. Gdynia is not an exception—in this case the factors which affect the walkability the most are accessibility and safety with other factors being not statistically significant in the model.

## 5. Conclusions

Political support for active mobility is growing for reasons such as land use planning [105], health, and improved mobility (quantitative and qualitative). Walking (and, to some extent, cycling) is regarded as the most democratic form of mobility [106].

It seems now that the primary concern of Gdynia should be a further increase in transportation accessibility and safety. Green areas, crossings and built infrastructure seem not to be perceived differently by people walking for the sake of walking and those who treat their walking trip as part of their everyday commute.

This way, the results are essential for urban planners and political decision-makers. They can be used to improve the understanding of what precisely affects the area being walkable.

While this itself can be often be quite easy to predict, our research, by the nature of the method of factor analysis being used, allows to see the interrelations between those factors. If two variables wounded up in the same factor and this factor positively affects the walkability of the area, as is the case with factor 1 (accessibility) and factor 5 (safety), making sure that at least one of those variables is positively perceived further increases the chances of other variables being positively perceived as well. This might allow to allocate the limited public funds in a more efficient away. At the same time, the results may be a basis or point of reference for other researchers analysing the subject of walkability. It is currently difficult to compare between cities that differ in size, the morphology of space, culture and socio-economic environment, or even climate. Different interview questionnaires are also used, depending on the purpose of the survey. Another difference that makes the direct comparison of the results impossible is the various methods of sampling respondents in the conducted research. In general, the lack of an integrated and consistent methodological framework in walkability research is evident [26, 107]. However, the above challenges should not limit pedestrian traffic research, which is crucial for cities implementing sustainability and resilience solutions. Our results can be a basis for a further research in this area as the methods applied are relatively universal.

Walking, regarded as a "slow mode", is particular and strongly influenced by the local context [108]. In general, "*walkability is affected by the design of the built environment and its features*" [24], but according to the detailed methodology and research approach [12, 109, 110] as well as to specific features of a given place (city), different results are obtained.

### 5.1. Limitations and further research

The limitations of this research were presented in part 2. Further research should be focused on the analysis of differences in walkability between areas of different density and different types of land use. It should include more variables connected to the built environment. Furthermore, a longitudinal study in which changes in other people's walking patterns would be analysed to separate the effect that the built environment has on mobility habits.

The authors call for developing a unified interview questionnaire to compare the results of studies on walking mobility in different cities.

## Supporting information

**S1 File.**
(DOC)

## Acknowledgments

The authors would like to thank Alicja Pawłowska (City Office of Gdynia), Wojciech Folejewski, Andrzej Ryński (ZDiZ Gdynia), Iwona Markesic, Paulina Szewczyk (Spatial Planning Bureau of the City of Gdynia), Weronika and Dean Edmunds for their support in preparing the paper.

## Author Contributions

**Conceptualization:** Marcin Wolek.

**Data curation:** Tomasz Czuba.

**Formal analysis:** Tomasz Czuba.

**Investigation:** Michal Suchanek.

**Methodology:** Marcin Wolek.

**Project administration:** Marcin Wolek.

**Resources:** Tomasz Czuba.

**Software:** Michal Suchanek.

**Validation:** Michal Suchanek.

**Visualization:** Marcin Wolek, Tomasz Czuba.

**Writing – original draft:** Marcin Wolek.

**Writing – review & editing:** Michal Suchanek.

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
