## [Decision Letter · Decision Letter 0]

23 Apr 2021

PONE-D-21-08683

Factors influencing walking trips. Evidence from Gdynia, Poland

PLOS ONE

Dear Dr. Wolek,

Thank you for submitting your manuscript to PLOS ONE. After careful consideration, we feel that it has merit but does not fully meet PLOS ONE’s publication criteria as it currently stands. Therefore, we invite you to submit a revised version of the manuscript that addresses the points raised during the review process.

We look forward to receiving your revised manuscript.

Kind regards,

Lubos Buzna, Ph.D

Academic Editor

PLOS ONE

Journal Requirements:

Please include additional information regarding the survey or questionnaire used in the study and ensure that you have provided sufficient details that others could replicate the analyses. For instance, if you developed a questionnaire as part of this study and it is not under a copyright more restrictive than CC-BY, please include a copy, in both the original language and English, as Supporting Information.

Thank you for stating the following in the Acknowledgments Section of your manuscript:

This research did not receive any specific grant from funding agencies in the public,

commercial, or not-for-profit sectors.

The authors would like to thank Alicja Pawłowska (City Office of Gdynia), Iwona Markesic,

Paulina Szewczyk (Spatial Planning Bureau of the City of Gdynia) and Dean Edmunds for their

support in preparing the paper.

Please ensure that you include a title page within your main document. You should list all authors and all affiliations as per our author instructions and clearly indicate the corresponding author.

We note you have included a table to which you do not refer in the text of your manuscript. Please ensure that you refer to Table 5 in your text; if accepted, production will need this reference to link the reader to the Table.

Please include captions for your Supporting Information files at the end of your manuscript, and update any in-text citations to match accordingly. Please see our Supporting Information guidelines for more information: http://journals.plos.org/plosone/s/supporting-information.

We note that Figure 2 in your submission contain map images which may be copyrighted. All PLOS content is published under the Creative Commons Attribution License (CC BY 4.0), which means that the manuscript, images, and Supporting Information files will be freely available online, and any third party is permitted to access, download, copy, distribute, and use these materials in any way, even commercially, with proper attribution. For these reasons, we cannot publish previously copyrighted maps or satellite images created using proprietary data, such as Google software (Google Maps, Street View, and Earth). For more information, see our copyright guidelines: http://journals.plos.org/plosone/s/licenses-and-copyright.

7a, You may seek permission from the original copyright holder of Figure 2 to publish the content specifically under the CC BY 4.0 license. 

7b, If you are unable to obtain permission from the original copyright holder to publish these figures under the CC BY 4.0 license or if the copyright holder’s requirements are incompatible with the CC BY 4.0 license, please either i) remove the figure or ii) supply a replacement figure that complies with the CC BY 4.0 license. Please check copyright information on all replacement figures and update the figure caption with source information. If applicable, please specify in the figure caption text when a figure is similar but not identical to the original image and is therefore for illustrative purposes only.

Reviewers' comments:

Reviewer's Responses to Questions

**Comments to the Author**

1. Is the manuscript technically sound, and do the data support the conclusions?

Reviewer #1: Yes

Reviewer #2: Partly

2. Has the statistical analysis been performed appropriately and rigorously? 

Reviewer #1: Yes

Reviewer #2: N/A

3. Have the authors made all data underlying the findings in their manuscript fully available?

Reviewer #1: Yes

Reviewer #2: Yes

4. Is the manuscript presented in an intelligible fashion and written in standard English?

Reviewer #1: Yes

Reviewer #2: Yes

5. Review Comments to the Author

Reviewer #1: Overall, this is a clear, concise, and well-written manuscript. References are relevant and have covered recent studies. The methods are generally appropriate. Overall, the results are clear and specific comments are as follows:

• Authors should further elaborate on what is the difference between the service accessibility as a part of Factor1 and accessibility to leisure activities as a part of Factor4? Why authors did not consider accessibility to leisure activities as a part of people’s assessment of accessibility and comfort?

• Authors claims that weather conditions also recorded on each day during the data collection period. Why these important data have not been considered as an influence factor in data modelling?

• In line 198, authors state “The overall evaluation of the pedestrian space in the centre of Gdynia is positive” that it is not clear which study they are referring to? It should be also mentioned here that following sentences in line 199 and 200 stand alone and there is no connection with the pervious part.

• Results in Table3, can be structured in a better way. Authors should further elaborate on presentation of sample description in Table3.

• In line 333 this sentence “The evaluation of time and distance of the walking journey is influenced by the perception of safety influences (Ralph et al., 2020)" can be removed as it has not been taken into account for evaluation of people’s perception of safety of walking and road traffic safety .this sentence has been also replicated in line 349.

Overall, this manuscript can be accepted after applying the requested revisions and proofreading.

Reviewer #2: - More suitable title should be selected for the article. Title should decrease to 10-12 words.

- The abstract should state briefly the purpose of the research, the principal results and major conclusions. An abstract is often presented separately from the article, so it must be able to stand alone.

- It is suggested to present the structure of the article at the end of the introduction.

- A flowchart should be added to the article to show the research methodology.

- The major defect of this study is the debate or Argument is not clear stated in the introduction session. Hence, the contribution is weak in this manuscript. I would suggest the author to enhance your theoretical discussion and arrives your debate or argument.

- More suitable title should be selected for the table 3 instead of “Sample characteristic (n=934).”.

- It is suggested to add articles entitled “Guo et al. Weather Impact on Passenger Flow of Rail Transit Lines”, “Habeeb and Talib Weli. Relationship of Smart Cities and Smart Tourism: An Overview” and “Abdulrazzaq et al. Traffic Congestion: Shift from Private Car to Public Transportation” to the literature review.

- It is suggested to compare the results of the present research with some similar studies which is done before.

- Page 16: the following paragraph is unclear, so please reorganize that:

“The third factor brings together the assessment of pedestrian crossings. Interestingly enough, all aspects of the crossings end up in the same factor: the waiting time, green light phases, traffic lights and safety aspects. In general, they are perceived rather poorly by the respondents – only 2% of respondents perceive pedestrian crossing waiting time as excellent.”

- Much more explanations and interpretations must be added for the Results, which are not enough.

- Please make sure your conclusions' section underscore the scientific value added of your paper, and/or the applicability of your findings/results, as indicated previously. Please revise your conclusion part into more details. Basically, you should enhance your contributions, limitations, underscore the scientific value added of your paper, and/or the applicability of your findings/results and future study in this session.

- “Notation” should be added to the article.

6. PLOS authors have the option to publish the peer review history of their article (what does this mean?). If published, this will include your full peer review and any attached files.

Reviewer #1: No

Reviewer #2: No

---

## [Author Response · Author response to Decision Letter 0]

23 May 2021

Editor 1 remark: 

OUR ANSWER:

The naming style was adjusted according to the requirements stated in: 

https://storage.googleapis.com/plos-published-prod/ba62/PLOSOne_formatting_sample_title_authors_affiliations.pdf?X-Goog-Algorithm=GOOG4-RSA-SHA256&X-Goog-Credential=wombat-sa%40plos-prod.iam.gserviceaccount.com%2F20210517%2Fauto%2Fstorage%2Fgoog4_request&X-Goog-Date=20210517T130608Z&X-Goog-Expires=3600&X-Goog-SignedHeaders=host&X-Goog-Signature=7c6221aa20a348ad3525e434920ec0203ab6bfbaeae39cd2b807ca291fa745b3607d83021b6de297f5e9d3f6bad1b7c6a0d199f67cf1041286a034a2af6588f171360b97f403506abd6a1f993cde1b9ec54c7927b5d780d08b7795f886b70b268d6833ee2b2e43da876e92d954e2b98fa79072fea6c43cd3a40096485e623bb8b64216bfa9ea9fc5432c931cb217fcebad2e0399a59dfea52c7c8450ddb662e1c770c8bfcc6fb804c3e05cf7d174093bf8d66183e813b41aa4be9cd0b77296fc03a8901d723e95acb8fca71d314d33268ed33ba88ec670ef8cf568ef81fe66a036064657dbeee0a3db21f08f2478a3e54c70976e8095f16060c46bcf56736774

EDITOR REMARK 2:

Please include additional information regarding the survey or questionnaire used in the study and ensure that you have provided sufficient details that others could replicate the analyses. For instance, if you developed a questionnaire as part of this study and it is not under a copyright more restrictive than CC-BY, please include a copy, in both the original language and English, as Supporting Information.

OUR RESPONSE:

The questionnaire in translated version (English) was submitted as an attachment. 

It was developed by the ZDiZ Gdynia (Road and Greenery Authority in Gdynia, a budgetary unit of the local self-government administration) in the framework of the EU-funded project – FLOW. Therefore it is publicly available as it was co-financed from the public (EU) sources. Moreover, we’ve got a permission from ZDiZ to use it for the purpose of our paper (please see thanks to the representatives of the ZDiZ and the City of Gdynia in the acknowledgements). 

EDITOR REAMRK 3:

Thank you for stating the following in the Acknowledgments Section of your manuscript:

This research did not receive any specific grant from funding agencies in the public,

commercial, or not-for-profit sectors.

The authors would like to thank Alicja Pawłowska (City Office of Gdynia), Iwona Markesic,

Paulina Szewczyk (Spatial Planning Bureau of the City of Gdynia) and Dean Edmunds for their

support in preparing the paper.

OUR RESPONSE:

Thank you very much for the comment. The funding information was deleted from the manuscript (in Acknowledgement part). 

EDITOR REMARK 4:

Please ensure that you include a title page within your main document. You should list all authors and all affiliations as per our author instructions and clearly indicate the corresponding author.

OUR RESPONSE:

The title page was included and all authors and all affiliations were added. 

EDITOR REMARK 5:

We note you have included a table to which you do not refer in the text of your manuscript. Please ensure that you refer to Table 5 in your text; if accepted, production will need this reference to link the reader to the Table.

OUR RESPONSE:

The reference in the text to Table 5 was added

EDITOR REMARK 6:

Please include captions for your Supporting Information files at the end of your manuscript, and update any in-text citations to match accordingly. Please see our Supporting Information guidelines for more information: http://journals.plos.org/plosone/s/supporting-information.

OUR RESPONSE:

All citations, including the new ones (suggested by the Reviewers) were implemented and updated. 

Captions: a link to the English version of the questionnaire was generated automatically. But we have added at the end of our manuscript a sentence: “A link to the English version of the questionnaire:”

EDITOR REMARK 7:

We note that Figure 2 in your submission contain map images which may be copyrighted. All PLOS content is published under the Creative Commons Attribution License (CC BY 4.0), which means that the manuscript, images, and Supporting Information files will be freely available online, and any third party is permitted to access, download, copy, distribute, and use these materials in any way, even commercially, with proper attribution. For these reasons, we cannot publish previously copyrighted maps or satellite images created using proprietary data, such as Google software (Google Maps, Street View, and Earth). [...]

OUR RESPONSE:

The previous version of the Figure being discussed (currently Figure 3) was removed. Instead, a new figure prepared by the Authors and based on:

1. the files obtained from the Spatial Planning Bureau of Gdynia

2. OpenStreetMap, 

which is supported with the respective citation under the figure. We hope that this time it is OK. If not, a written permission from the Spatial Planning Bureau could be provided.

We’ve read the link:

https://wiki.osmfoundation.org/wiki/Licence where the statement is:

“Data extracted from OpenStreetMap after September 2012 is licensed on terms of the Open Database License, "ODbL" 1.0, previously it was licensed CC-BY-SA 2.0”. 

Therefore the source under the picture is:

Source: self-study based on the (City of Gdynia, 2019) and OpenStreetMap

We hope that this explanation and a new version of the picture fulfils your restrictive rules. If not, we are ready to revise our picture to avoid a collision with the PLOS ONE rules. 

1ST REVIEWER REMARK NR 1: 

Authors should further elaborate on what is the difference between the service accessibility as a part of Factor1 and accessibility to leisure activities as a part of Factor4? Why authors did not consider accessibility to leisure activities as a part of people’s assessment of accessibility and comfort?

OUR RESPONSE:

Thank you for the comment. The factors represent the result of exploratory factor analysis. As opposed to the confirmatory factor analysis, this means that there were no a priori assumptions about the structure or variability of the factors. Interestingly enough, accessibility of leisure activities turned out to be grouped together with the characteristics of the pavements and their quality. This might indicate that the fact that the pedestrian can participate in leisure activities during the trip further improves his/her perception of the quality of pavement itself. A possibility of the interpretation of the result was included in the paragraph in the description of the fourth factor.

1ST REVIEWER REMARK NR 2: 

Authors claims that weather conditions also recorded on each day during the data collection period. Why these important data have not been considered as an influence factor in data modelling?

OUR RESPONSE:

Thank you for the comment. In the section 3.3. we explained that “Weather conditions were recorded on each day during the data collection period but they had little impact on pedestrian traffic. Therefore it was assumed that weather conditions had no substantial impact on the research.” The sampling was conducted regardless of weather conditions. We also noted that the weather did not affect the results of the qualitative research, collected through a personal interview questionnaire.

1ST REVIEWER REMARK NR 3:

In line 198, authors state “The overall evaluation of the pedestrian space in the centre of Gdynia is positive” that it is not clear which study they are referring to? It should be also mentioned here that following sentences in line 199 and 200 stand alone and there is no connection with the pervious part.

OUR RESPONSE:

Thank you for this comment. After revision, and internal discussion we’ve decided to exclude this text as it was taken from internal research of the City of Gdynia administration but was not directly confirmed by the respective research that we could cite in the paper.

1ST REVIEWER REMARK NR 4:

Results in Table3, can be structured in a better way. Authors should further elaborate on presentation of sample description in Table3.

OUR RESPONSE:

We agree with this remark. Therefore, the title of the Table 3 was changed from “Sample characteristic (n=934)” to “ Basic characteristics of the respondents”. 

The table was structured in a better way, according to the Reviewer’s remark. A more complex description of the results was provided in the text related to the Table 3. 

1ST REVIEWER REMARK NR 5:

In line 333 this sentence “The evaluation of time and distance of the walking journey is influenced by the perception of safety influences (Ralph et al., 2020)" can be removed as it has not been taken into account for evaluation of people’s perception of safety of walking and road traffic safety .this sentence has been also replicated in line 349.

OUR RESPONSE:

We agree with this remark and the sentence in line 333 was deleted. 

2ND REVIEWER REMARK NR 1:

More suitable title should be selected for the article. Title should decrease to 10-12 words.

OUR RESPONSE:

After discussion and several attempts, we have decided to leave the title in the form proposed in the first version of our manuscript. The title consists of 8 words and in our opinion describes the topic quite precisely. Its length is even shorter than Reviewer suggests. (by the way we agree that title should not exceed 10-12 words). 

2ND REVIEWER REMARK NR 2: 

The abstract should state briefly the purpose of the research, the principal results and major conclusions. An abstract is often presented separately from the article, so it must be able to stand alone.

OUR RESPONSE:

We agree with this comment, as this remark perfectly describes basic parameters of the perfect abstract. The abstract was rewritten, to the form that states the goal of the research (“This paper aims to identify the main factors affecting the walkability of the city centre of Gdynia (Poland)”) and main findings (“The results show that only two of the analysed factors are significant, namely accessibility and safety. Both are extensive categories and include many subcomponents that are influential among different groups of citizens. Our research also confirms that walkability is a city-specific issue that is influenced by many local factors”). 

2ND REVIEWER REMARK NR 3:

It is suggested to present the structure of the article at the end of the introduction.

OUR RESPONSE:

We agree with this remark, therefore the structure was described in the form of separated subchapter (1.2) Structure of the paper.

2ND REVIEWER REMARK NR 4:

A flowchart should be added to the article to show the research methodology.

OUR RESPONSE:

We agree, and the picture presenting research methodology and the process was added in the section 3 (Figure 1. The research process). 

2ND REVIEWER REMARK NR 5:

The major defect of this study is the debate or Argument is not clear stated in the introduction session. Hence, the contribution is weak in this manuscript. I would suggest the author to enhance your theoretical discussion and arrives your debate or argument.

OUR RESPONSE:

According to this remark following actions were implemented:

1. The structure of the section 4 was improved (division into 3 sub-sections, development of a new sub-section 4.3. Discussion).

2. Improvement of the Section 4.1. with a new text. Two paragraphs were added describing the choice of the factor analysis method and the process of it being carried out (section 4).

3. Conclusions (section 5) were developed and enriched with a new text.

Also, section 4.1. and 4.2. contains part of the discussion as we wanted to present results supported with the discussion immediately. 

2ND REVIEWER REMARK NR 6:

More suitable title should be selected for the table 3 instead of “Sample characteristic (n=934).”.

OUR RESPONSE:

The title was changed to “Basic characteristics of the respondents”. Moreover, the table was structured in a better way.

2ND REVIEWER REMARK NR 7:

It is suggested to add articles entitled “Guo et al. Weather Impact on Passenger Flow of Rail Transit Lines”, “Habeeb and Talib Weli. Relationship of Smart Cities and Smart Tourism: An Overview” and “Abdulrazzaq et al. Traffic Congestion: Shift from Private Car to Public Transportation” to the literature review.

OUR RESPONSE:

Thank you very much for the valuable suggestions. After thorough analysis we decided to use all of the suggested papers. 

2ND REVIEWER REMARK NR 8:

It is suggested to compare the results of the present research with some similar studies which is done before.

OUR RESPONSE:

Thank you for this remark. This is the first study of this kind in the city of Gdynia. Relevant explanation was put in the text. Previous studies focused on evaluation of the scale of pedestrian traffic in the city centre. A discussion with the results obtained in other cities was provided and enriched in the section 4.

2ND REVIEWER REMARK NR 9:

Page 16: the following paragraph is unclear, so please reorganize that:

“The third factor brings together the assessment of pedestrian crossings. Interestingly enough, all aspects of the crossings end up in the same factor: the waiting time, green light phases, traffic lights and safety aspects. In general, they are perceived rather poorly by the respondents – only 2% of respondents perceive pedestrian crossing waiting time as excellent.”

OUR RESPONSE:

Thank you for this remark, as a result of it, two paragraphs were added describing the choice of the factor analysis method and the process of it being carried out. This has been rephrased, and now it is: “The third factor groups all the variables connected with the assessment of pedestrian crossings. Interestingly enough, it turned out that all the variables connected with the pedestrian crossing behave in a similar fashion – if the respondent assessed one of the aspects positively, it was quite likely that he assessed all of the positively. These aspects include: the waiting time, green light phases, traffic lights and safety aspects”. 

2ND REVIEWER REMARK NR 10:

Much more explanations and interpretations must be added for the Results, which are not enough.

OUR RESPONSE:

Thank you for this comment. As a result, we have enriched the text in the section 4 and six new paragraphs were added to provide more complex explanations of the results and the research procedure (especially in case of factor analysis). Moreover, one larger paragraph was added to the Section 5 (Conclusions). 

2ND REVIEWER REMARK NR 11:

Please make sure your conclusions' section underscore the scientific value added of your paper, and/or the applicability of your findings/results, as indicated previously. Please revise your conclusion part into more details. Basically, you should enhance your contributions, limitations, underscore the scientific value added of your paper, and/or the applicability of your findings/results and future study in this session.

OUR RESPONSE:

Thank you for this comment. A paragraph regarding further applications and limitations has been significantly extended (in the section 5 Conclusions). Moreover, we still have section on “Limitations and further research”, that follows the fifth section (Conclusions).

2ND REVIEWER REMARK NR 12:

“Notation” should be added to the article.

OUR RESPONSE:

Thank you for this comment. We understand it according to the definition of “notation” is provided by Merriam Webster dictionary ( https://www.merriam-webster.com/dictionary/notation ). Therefore, we have implemented this remark in the Table 1, improving notations of values.

---

## [Decision Letter · Decision Letter 1]

7 Jul 2021

Factors influencing walking trips. Evidence from Gdynia, Poland

PONE-D-21-08683R1

Dear Dr. Wolek,

We’re pleased to inform you that your manuscript has been judged scientifically suitable for publication and will be formally accepted for publication once it meets all outstanding technical requirements.

Kind regards,

Lubos Buzna, Ph.D

Academic Editor

PLOS ONE

Additional Editor Comments (optional):

Reviewers' comments:

Reviewer's Responses to Questions

**Comments to the Author**

1. If the authors have adequately addressed your comments raised in a previous round of review and you feel that this manuscript is now acceptable for publication, you may indicate that here to bypass the “Comments to the Author” section, enter your conflict of interest statement in the “Confidential to Editor” section, and submit your "Accept" recommendation.

Reviewer #1: All comments have been addressed

Reviewer #2: All comments have been addressed

2. Is the manuscript technically sound, and do the data support the conclusions?

Reviewer #1: Yes

Reviewer #2: Yes

3. Has the statistical analysis been performed appropriately and rigorously? 

Reviewer #1: Yes

Reviewer #2: Yes

4. Have the authors made all data underlying the findings in their manuscript fully available?

Reviewer #1: No

Reviewer #2: Yes

5. Is the manuscript presented in an intelligible fashion and written in standard English?

Reviewer #1: Yes

Reviewer #2: Yes

6. Review Comments to the Author

Reviewer #1: The newly added sentences in the revised manuscript are quite often very long. So, is very difficult to understand the content with a fast reading of the paper(e.g. line 410-415, line 562- 571). Thus the manuscript needs to be proofread by a proofreader before it can be published.

Reviewer #2: (No Response)

7. PLOS authors have the option to publish the peer review history of their article (what does this mean?). If published, this will include your full peer review and any attached files.

Reviewer #1: No

Reviewer #2: No

---

## [Editor Report · Acceptance letter]

23 Jul 2021

PONE-D-21-08683R1 

Factors influencing walking trips. Evidence from Gdynia, Poland 

Dear Dr. Wolek:

I'm pleased to inform you that your manuscript has been deemed suitable for publication in PLOS ONE. Congratulations! Your manuscript is now with our production department. 

Kind regards, 

on behalf of

Prof. Lubos Buzna 

Academic Editor

PLOS ONE